# Effective Self-Supervised Transformers For Sparse Time Series Data

## Abstract

Electronic health records (EHRs) recorded in hospital settings such as intensive care units (ICUs) typically contain a wide range of numeric time series data that is characterized by high sparsity and irregular observations. Self-supervised Transformer architectures have shown outstanding performance in a variety of structured tasks in natural language processing and computer vision. However, the sparse irregular time series nature of ICU EHR data poses challenges for the application of transformers that have not been widely explored. One of the major challenges is the quadratic scaling of self-attention layers that can significantly limit the input sequence length. In this work, we introduce TESS, **T**ransformers for **E**HR data with **S**elf **S**upervised learning, a self-supervised Transformer-based architecture designed to extract robust representations from EHR data. We propose the application of input binning to aggregate the time series inputs and sparsity information into a regular sequence with fixed length, enabling the training of larger and deeper Transformers. We demonstrate that significant compression of ICU EHR data is possible without sacrificing useful information, likely due to the highly correlated nature of observations in small time bins. We then introduce self-supervised prediction tasks that provide rich and informative signals for model pre-training. TESS outperforms state-of-the-art deep learning models on multiple downstream tasks from the MIMIC-IV and PhysioNet-2012 ICU EHR datasets.

## 1 Introduction

Electronic health record (EHR) data collected in the hospital contains an immense amount of information about patients. This data typically comes in the form of vital sign measurements, lab results, and diagnoses/treatments. Patients in an Intensive Care Unit (ICU) are particularly heavily monitored, with frequent vital sign observations and diagnostic tests. The resulting multivariate numeric time series is high-dimensional, sparse, and irregularly distributed across time, making it challenging to apply standard time series analysis methods that are primarily designed for densely sampled data. These challenges are not unique to health care, and data with such characteristics commonly arises in fields such as finance, banking, and e-commerce (Cao et al., 2021; Gómez-Losada & Duch-Brown, 2019; Zhang et al., 2015).

Good models of clinical outcomes need to extract predictive signal from the values, frequencies and missingness patterns from such data. Hand-crafting such features is a non-trivial and time-consuming task, which has led to the exploration of deep learning for problems arising in healthcare. However, when the labels are noisy and scarce, such methods too are susceptible to overfitting. Self-supervised learning (SSL) (Chopra et al., 2005; Caron et al., 2021), has risen in popularity as a tool to reduce the dependence of representation learning on large amounts of labelled data. SSL relies on the premise that domain experts have prior knowledge about the patterns in high-dimensional data; by translating this domain knowledge into *pseudo-tasks*, practitioners can ensure that this knowledge is transferred to representation learning models prior to fine-

tuning them with labelled data. The premise of SSL is attractive for EHR data, where few positive samples might be observed for a desired outcome and privacy limitations can prevent the collection of larger labelled datasets (Krishnan et al., 2022; Bak et al., 2022).

Methods for SSL are often applied to Transformers (Vaswani et al., 2017), which have proven to be an effective neural architecture for finding useful patterns across a variety of different domains. Self-supervised Transformer models currently produce state-of-the-art results in natural language processing (NLP) (Brown et al., 2020), computational histopathology (Chen & Krishnan, 2021; Chen et al., 2022), computer vision (He et al., 2022), and cross-modal learning (Radford et al., 2021). On numeric EHR data (Li et al., 2021; Ren et al., 2021; Tipirneni & Reddy, 2022) however, there remain many open questions regarding good self-supervised tasks and whether SSL and Transformers can achieve the same level of success as in other fields.

In this work, we present TESS, an approach for self-supervised training of Transformers on ICU EHR data that produces representations which generalise well to different downstream tasks of interest. Prior work along this vein has used embedded input sequences that have one sequence element for every patient event (Li et al., 2021; Tipirneni & Reddy, 2022). This is limited in scalability as patients can have hundreds of events in a relatively short period of time, while memory and runtime complexity of self-attention layers scales quadratically with input length. More efficient attention layers have been proposed (Wang et al., 2020), but state-of-the-art Transformer models still generally use quadratic attention. Training large models with this input representation consequently requires significant hardware resources or aggressive input truncation, which can negatively impact accuracy.

**Contributions:** To address the limitations discussed above, we first propose an application of time binning to compress the input. We observe that increasing time resolution beyond a certain point does not improve model performance, likely due to consecutive measurements of the same event within a small time window being highly correlated. By aggregating events within time bins and including auxiliary data describing the input sparsity structure, we can significantly compress the input without substantial loss of useful information. Each bin is projected using a multi-layer perceptron (MLP) to a given embedding dimension, and combined with an embedded representation of the time period the bin represents. This shorter and denser input allows us to train larger and deeper Transformer models, which in turn leads to better representations.

Next, we propose an SSL approach to train TESS. Measurements that clinicians choose to take reflect their understanding of a patient's condition and related treatment strategies. Consequently, the missingness patterns of different events contain predictive signals of interest for a variety of different tasks (Lipton et al., 2016) since they represent a clinician's unobserved *lack of intent* to treat or measure the value in question. A good representation of a patient's state should be aware of both the past state that the patient transitioned from, as well as the future states they might evolve into. We construct SSL tasks by predicting a combination of both missingness masks and event values. Since adjacent measurements of the same event are likely to be highly correlated, we introduce a masked event type dropout scheme that encourages the model to design representations that pull information from other time-bins and events rather than using simple interpolation.

We evaluate TESS on the MIMIC-IV (Johnson et al., 2022) and PhysioNet-2012 (Silva et al., 2012) datasets, showing that it outperforms state-of-the-art baselines on multiple downstream tasks such as mortality prediction and phenotype classification. We also evaluate the efficacy of self-supervised learning with TESS, showing that it learns to produce a good representation of patients that can be fine-tuned effectively with only a small amount of labelled data.

## 2 RELATED WORK

**Deep learning for sparse irregular time series.** A variety of neural network models have been proposed for supervised learning on sparse irregular time series data. Most are based on recurrent neural networks (Hochreiter & Schmidhuber, 1997; Cho et al., 2014) that expect regularly sampled inputs with-

out missingness, so modifications are required to account for sparse data. One simple approach is to impute missing values based on previous values or overall statistics. mTANs (Shukla & Marlin, 2021) use a more advanced attention-based interpolation approach to produce a regular input for an RNN model. Architectural modifications can also be added to allow RNNs to adapt their hidden state appropriately when inputs are missing, as in CT-GRU (Mozer et al., 2017) and GRU-D (Che et al., 2018). Another line of research uses differential equations to model underlying continuous processes that are related to irregularly sampled inputs (Rubanova et al., 2019; Lechner & Hasani, 2020; Kidger et al., 2020), but these approaches require the use of differential equation solvers during training and inference, usually making them slower than ordinary neural networks (Shukla & Marlin, 2021). More recently, Raindrop (Zhang et al., 2022) has applied graph neural networks to aggregate observation embeddings, achieving state-of-the-art results.

The success of Transformer models in NLP makes them an attractive candidate for other tasks involving sequential data. Many Transformer-based models have been proposed for regular time series data (Wen et al., 2022), but there are fewer models that extend them to sparse irregular time series. RAPT (Ren et al., 2021) introduces a modified time-aware attention mechanism to deal with irregular inputs. STraTS (Tipirneni & Reddy, 2022) instead embeds every individual observation as triplets of time, variable and value using MLPs, then passes this sequence of observations to a Transformer. This approach suffers from the limitation that Transformer memory usage is quadratic in sequence length, meaning that the amount of observed data is practically limited and the method must use very small Transformers (the proposed model only uses two Transformer layers and embeddings of length 50). A different approach, Hi-BEHRT (Li et al., 2021), uses a hierarchical Transformer architecture to be able to process longer input sequences of individual observations.

In contrast, our approach addresses this limitation by creating a binned time series input representation. This approach is loosely inspired by recent work in computer vision, which showed that processing patches of images before passing them to a Transformer achieves excellent results (Dosovitskiy et al., 2021), suggesting that patching can be used to adapt Transformers to new data modalities.

Previous works have applied transformers to model longitudinal EHR data by using the sets of diagnostic codes applied at hospital or doctor visits as their inputs (Rasmy et al., 2021; Li et al., 2020; Zhang et al., 2020). However, these models are not applicable to the kinds of data we investigate here, since their inputs do not contain numeric measurements and their temporal resolution is limited to one observation per visit.

**Self-supervised learning.** SSL has become an important framework to enable learning useful representations from data without relying on labels. SSL with Transformers has driven recent advances in NLP and computer vision (e.g. Devlin et al. (2019); Dosovitskiy et al. (2021)), and has clear potential to advance other fields. Different approaches for SSL with numeric EHR data are explored in McDermott et al. (2021), but these are applied to a relatively basic GRU model and do not claim to achieve state-of-the-art results. mTANs use a similar approach to SSL based on reconstructing inputs, incorporating a VAE loss into their training procedure, but without a distinct pre-training stage. SSL is used with Transformers in Hi-BEHRT, which applies BYOL (Grill et al., 2020) to augmentations of EHR time series data. STraTs uses masked value prediction for SSL with Transformers, while RAPT additionally uses a reasonability check to identify corrupted sequences and a contrastive patient similarity task. Our SSL approach instead adds a presence/absence prediction task, which captures meaningful priors of clinicians regarding a patient's state.

## 3 TRANSFORMER FOR EHR DATA WITH SELF SUPERVISED LEARNING

We consider a sparse irregular time series dataset of the form $\mathcal{D} = \{(\boldsymbol{s}^p, \boldsymbol{W}^p, y^p)\}_{p=1}^N$, where each patient stay $p$ is associated with a set of outcomes $y^p$, a vector of static inputs $\boldsymbol{s}^p \in \mathbb{R}^{n_{\text{static}}}$ and a sequence of events $\boldsymbol{W}^p = (w_1^p, w_2^p, \cdots, w_{n_p}^p)$ of variable length, $n_p$. Each event $w_j^p$ contains a triplet $(f_j^p, t_j^p, v_j^p)$ representing event-type, time since start of stay and value (if applicable); for example, [heart_rate, 5.32 days, 41bpm]. Static variables do not change during the stay and contain data such as age, gender, and so on.

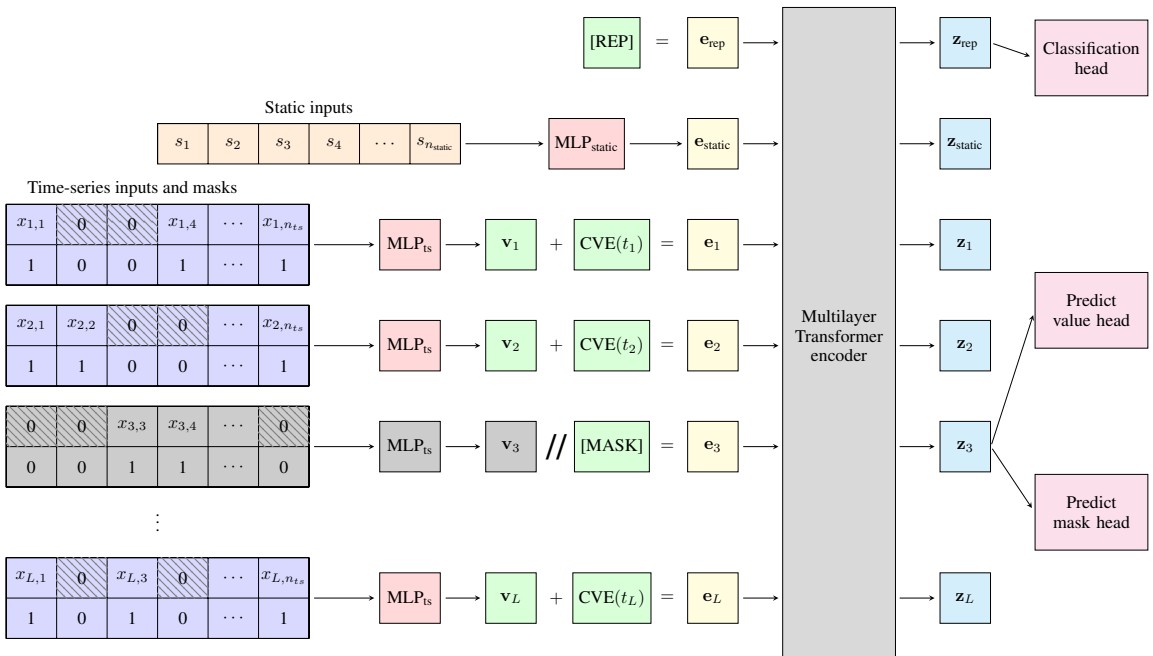

Figure 1: Architecture diagram for a single patient stay. Binned time series events $x$ and corresponding masks are passed through the $\text{MLP}_{\text{ts}}$ encoder to produce $d$-dimensional representations $\mathbf{v}$. These representations are combined with the time embedding $\text{CVE}(t)$ for each bin. Similarly, static patient data $\mathbf{s}$ is passed through the $\text{MLP}_{\text{static}}$ encoder and concatenated with bin representations to form the inputs $\mathbf{e}$ to the Transformer. During pre-training, bins are masked with a learned [MASK] embedding and then reconstructed via mask and value heads. For downstream fine-tuning, a patient representation [REP] is added to the input sequence and the corresponding Transformer output $\mathbf{z}_{\text{rep}}$ is used to attach the downstream task head.

In the following section, we present the construction of the input representation for TESS from the full event sequence for each patient stay and omit the corresponding index $p$ for notational simplicity. Then, we outline the architecture, SSL pre-training, and fine-tuning approach.

## 3.1 MODEL ARCHITECTURE

The first step of our input processing splits the full sequence of patient events, $\boldsymbol{W}$, into $L$ time bins of equal length. For each patient stay, we define a binned input matrix $\boldsymbol{x} \in \mathbb{R}^{L \times n_{\text{ts}}}$ where the element $x_{l,i}$ contains a single value representing an aggregation of all observed values of event-type $i$ in a given time bin $l$, and $n_{\text{ts}}$ is the number of unique event-types selected as inputs to the model. We can express $x_{l,i}$ as

$$x_{l,i} = g(\mathbf{S}_{l,i}) \quad \text{where } \mathbf{S}_{l,i} = \{w_j | f_j = i \wedge (l-1)b < t_j \le bl\} \tag{1}$$

and where $b = t_{n_p}/L$ is the bin size for the patient stay, and $g$ is an aggregation function. Choices for the aggregation function include the mean value of events, the maximum or minimum, or the last value observed in the time bin. Elements of $\boldsymbol{x}$ with no observations in the corresponding time bin are set to $0$. We also build a mask tensor $\boldsymbol{m} \in \mathbb{R}^{L \times n_{\text{ts}}}$ where $m_{l,i}$ is set to 1 if there are observations for event-type $i$ in time bin $l$, and 0 otherwise. Passing the presence/absence of events to the model provides useful information on the types of analyses and treatments that clinicians have selected for the patient, as well as allowing the model to distinguish between a measured zero value in $\boldsymbol{x}$ and a missing value. By adapting the number of bins

$L$, this input representation allows us to effectively control the trade-off between computational complexity and granularity of event information. In comparison with the input representation used by STraTS, this representation reduces the computational complexity from $O(n_p^2)$ to $O(L^2)$ in terms of time an memory, where generally, $L \ll n_p$. We also aim to develop an SSL approach that can be effectively applied to raw input data with minimal processing, as feature engineering can be highly time consuming and require extensive domain expertise, and so we only consider basic aggregation functions.

A time series encoder $\text{MLP}_{\text{ts}}$ is used to convert $\boldsymbol{x}_{l,:}$ and $\boldsymbol{m}_{l,:}$ input representations for a given time bin $l$ into a fixed-size embedding of dimension $d$, which we denote as $\mathbf{v}_l \in \mathbb{R}^d$. The encoder is a fully connected feed-forward neural network with ReLU activation and batch normalization between layers and is shared for all time bins. The static data input $\boldsymbol{s}$ is similarly passed through a separate encoder $\text{MLP}_{\text{static}}$ to produce an encoding with the same dimensionality $\mathbf{v}_{\text{static}} \in \mathbb{R}^d$.

Since the overall length of time represented in the input bins can vary from patient to patient, only using the positional encodings would discard potentially useful information about the time scale. To incorporate this information, our model learns embeddings calculated from the continuous time values representing each time bin. We use the continuous value embedding (CVE) approach proposed in Tipirneni & Reddy (2022), which passes each time value through a fully connected feed-forward neural network with one hidden layer of size $\sqrt{d}$ and a tanh activation, followed by an output layer that produces a time embedding in $\mathbb{R}^d$. In addition to incorporating continuous time information, the neural network is able to learn an embedding function that is well adapted to the data. These embeddings are then added to the output of the time series encoder to produce a final embedding for each time bin. The time value for time bin $l$ is $t_l = bl$, the difference between the bin end time and start of the patient's stay, and represents the (fractional) number of days that have passed since the start of the stay.

In addition to time bin embeddings, we also prepend the static data embedding to the input sequence as well as the learned patient representation token [REP]. Incorporating the static variables into the input rather than via late fusion as in recent models (Tipirneni & Reddy, 2022; Zhang et al., 2022) enables the Transformer to fully leverage this information in every self-attention block. Static variables such as age and sex provide critical prior information on the patient that can considerably influence both treatment strategies and outcomes, so early fusion is beneficial here. The patient representation token queries a summary of relevant patient information, and the corresponding Transformer output embedding $\mathbf{z}_{\text{rep}}$ is used for downstream classification tasks. To summarize, the input embedding tensor $\boldsymbol{e} \in \mathbb{R}^{(L+2) \times d}$ is constructed according to:

$$\boldsymbol{e} = [[\text{REP}], \text{MLP}_{\text{static}}(\boldsymbol{s}), \text{MLP}_{\text{ts}}([\boldsymbol{x}_{1,:}; \boldsymbol{m}_{1,:}]) + \text{CVE}(t_1), \ldots, \text{MLP}_{\text{ts}}([\boldsymbol{x}_{L,:}; \boldsymbol{m}_{L,:}]) + \text{CVE}(t_L)]^T \quad (2)$$

Our approach of adding heterogeneous data by extending the input sequence makes it easy to incorporate other data sources and modalities. For instance, patches or other representations of medical images could be appended to the sequence along with learned embeddings representing their position, time, and modality. This is an especially promising possibility for health data, where medical images and doctors' notes are often available in addition to numeric measurements (Soenksen et al., 2022). This is not a straightforward task for most other models; for example, mTANs and Raindrop are defined explicitly for time series input and XGBoost for tabular inputs which limits their ability to incorporate other modalities such as image and text data. Similarly, STraTS proposes a late fusion of the tabular data which limits its ability to extract useful information through self-attention layers.

The input embeddings tensor is passed through a multi-layer Transformer encoder to produce a representation sequence $\boldsymbol{z} \in \mathbb{R}^{(L+2) \times d}$. We use the standard Transformer blocks as in Vaswani et al. (2017) with two modifications: we move normalization inside the residual connections as described in Xiong et al. (2020) and use ScaleNorm instead of LayerNorm as in Nguyen & Salazar (2019) to make training more stable. Dropout is used on feed-forward and attention connections. A full diagram of our model architecture is shown in Figure 1.

### 3.2 TRAINING

The model is trained in two phases: self-supervised pre-training, then fine-tuning on downstream tasks.

**SSL pre-training:** During pre-training, we adopt the mask-predict approach that has been highly successful in the NLP and computer vision domains (Devlin et al., 2019; He et al., 2022), and repeatedly mask time bin input embeddings $e_l$ with a learned [MASK] embedding. Bins are chosen uniformly at random for masking, and the number of masked bins at each training step is set as a hyperparameter. We design a self-supervised task focused on predicting both the presence/absence of an event and its value. For every masked bin $l$, the corresponding Transformer output $z_l$ is passed to mask and value prediction heads to produce the predictions $\hat{y}_l^{\text{mask}} \in \mathbb{R}^{n_{\text{ts}}}$ and $\hat{y}_l^{\text{value}} \in \mathbb{R}^{n_{\text{ts}}}$. The mask head predicts which events occurred in the time bin, and the value head predicts the value of observed events. These predictions are then compared with the actual values using the cross entropy and squared error losses. The pre-training loss for a single time bin $l$ is:

$$\mathcal{L} = \frac{1}{n_{\text{ts}}} \sum_{i=1}^{n_{\text{ts}}} \left[ m_{l,i} \left( \hat{y}_{l,i}^{\text{value}} - x_{l,i} \right)^2 - \alpha \left[ m_{l,i} \log(\hat{y}_{l,i}^{\text{mask}}) + (1 - m_{l,i}) \log(1 - \hat{y}_{l,i}^{\text{mask}}) \right] \right] \tag{3}$$

where $\alpha$ is a hyperparameter that controls the contribution of each task. We also introduce a novel masked event dropout scheme. For each event-type $i$ having $m_{l,i} = 1$ in at least one masked bin, all values of this event-type are dropped from the inputs by setting their values to 0 across all time bins with some fixed probability. Patient observations such as labs and vitals are typically highly correlated within a short time frame, so this dropout scheme prevents the model from learning to reconstruct events by simply interpolating from nearby time bins. The loss function and other inputs are unaffected.

**Fine-tuning:** During fine-tuning, we use the patient representation $\mathbf{z}_{\text{rep}}$ produced by the Transformer from the [REP] input and attach heads tailored to the downstream tasks. Note that the [REP] embedding value is learned at this stage, since its output is not used during pre-training. For the tasks explored in this paper we use MLP classification heads with sigmoid output and binary cross entropy loss.

## 4 EXPERIMENTS

We evaluate our proposed model on two widely used EHR datasets: MIMIC-IV (Johnson et al., 2022) and the PhysioNet/CinC Challenge 2012 (Silva et al., 2012). In this section, we present our data preprocessing steps, our experimental design and the results of evaluating our model performance compared to the leading baselines. We also present experiments to demonstrate the quality of the learned representation and conduct an ablation study to evaluate the impact of the components of our approach. We consider the following tasks to evaluate our models:

**MIMIC-IV** (Johnson et al., 2022) is a publicly available dataset that contains retrospective, deidentified data of patients admitted to the ICU or the emergency department at the Beth Israel Deaconess Medical Center between 2008 and 2019. This dataset contains data of various modalities such as time series data, static tabular data, and medical images from $53,150$ patients with $69,211$ admissions. Our dataset consists of ICU stays from MIMIC-IV along with related targets. Following Harutyunyan et al. (2019), we define mortality prediction and phenotype classification targets. We exclude patients below 18 years of age and patients with no chart or lab events recorded during the stay. Unlike Harutyunyan et al. (2019), we do not exclude patients with multiple ICU stays or transfers between ICU units during their stay. This results in a larger dataset that more closely mimics the practical use of a machine learning system in a hospital setting. The mortality prediction task uses the first $48$ hours of the patient stay as the input time window, predicting whether death occurs later during the hospital stay and has $13\%$ positive instances. Patients with stays of less than $48$ hours and patients with no recorded events before $48$ hours are excluded from this task. The phenotype classification task uses the entire ICU stay as the input time window and uses a multi-label classification target, predicting $25$ common hospital diagnoses. Details are provided in Appendix D. We include all input

Table 1: Performance on MIMIC-IV mortality and phenotyping, and PhysioNet-2012 mortality tasks. Phenotyping metrics are macro-averaged. Multiple top scores are bolded when the difference between them is not statistically significant according to the Wilcoxon signed rank test.

| Model | MIMIC-IV | | | | PhysioNet-2012 | |
| | Mortality | | Phenotyping | | Mortality | |
| | ROC-AUC | PR-AUC | ROC-AUC | PR-AUC | ROC-AUC | PR-AUC |
|---|---|---|---|---|---|---|
| XGBoost | $0.886 \pm 0.003$ | $0.593 \pm 0.004$ | $\mathbf{0.8287 \pm 0.0003}$ | $\mathbf{0.5891 \pm 0.0007}$ | $0.865 \pm 0.001$ | $0.531 \pm 0.01$ |
| LSTM | $0.881 \pm 0.001$ | $0.533 \pm 0.006$ | $0.7564 \pm 0.0002$ | $0.4467 \pm 0.0004$ | $0.848 \pm 0.002$ | $0.494 \pm 0.002$ |
| mTAND | $0.864 \pm 0.002$ | $0.540 \pm 0.007$ | $0.812 \pm 0.001$ | $0.553 \pm 0.003$ | $0.857 \pm 0.001$ | $0.515 \pm 0.007$ |
| STraTS | $0.882 \pm 0.004$ | $0.552 \pm 0.013$ | $0.8196 \pm 0.0008$ | $0.565 \pm 0.002$ | $0.852 \pm 0.008$ | $0.527 \pm 0.02$ |
| Raindrop | $0.878 \pm 0.001$ | $0.546 \pm 0.002$ | $0.824 \pm 0.001$ | $0.577 \pm 0.003$ | $0.838 \pm 0.01$ | $0.479 \pm 0.02$ |
| TE(SS-) | $0.878 \pm 0.004$ | $0.552 \pm 0.007$ | $0.818 \pm 0.0008$ | $0.562 \pm 0.002$ | $0.854 \pm 0.01$ | $0.522 \pm 0.02$ |
| TESS | $\mathbf{0.895 \pm 0.002}$ | $\mathbf{0.607 \pm 0.003}$ | $\mathbf{0.829 \pm 0.001}$ | $0.588 \pm 0.002$ | $\mathbf{0.871 \pm 0.004}$ | $\mathbf{0.561 \pm 0.007}$ |

variables used in Harutyunyan et al. (2019) as well as a number of static variables and all chart and lab events that are observed in more than $50\%$ of ICU stays. This substantially increases the set of variables, and provides a rich input signal to the model. The variables are listed in Appendix D. We use a patient-level $70\%:15\%:15\%$ split between the training, validation, and test sets.

**PhysioNet-2012** (Silva et al., 2012) is a standardized dataset with the task of predicting in-hospital mortality after the first $48$ hours of patient stays in the ICU, where $14\%$ of mortality labels are positive. The dataset consists of $12,000$ ICU stays with $42$ different variables including $37$ time series event-types. The details of the dataset, including data statistics, are provided in Silva et al. (2012). We use the torchtime (Darke et al., 2022) data library to preprocess the data in a standard way and to split the dataset into training, validation and test sets ($70\% : 15\% : 15\%$).

**Implementation:** We apply zero-mean and unit-standard deviation normalization to all inputs in $x$. We also clip outliers using a threshold of three median absolute deviations from the median. These steps allow for stable training without clinical knowledge of normal variable ranges. For all models, when doing supervised training, we use binary cross entropy loss. We weight positive and negative instances according to the target positive fraction so that they receive equal weight in the loss. TESS is trained with $L = 32$ time steps, $\alpha = 0.2$, batch size of $512$, and embedding dimension $d = 768$. Events in each time bin are aggregated by taking the last observed value. We perform self-supervised pre-training for 200 epochs using AdamW (Loshchilov & Hutter, 2017) with learning rate $10^{-4}$ and weight decay $10^{-6}$. Linear warmup is applied for the first $1000$ steps and the learning rate is reduced by a factor of 10 after 100 epochs. During pre-training, we apply dropout to the Transformer attention and feedforward layers with probability 0.2. We mask one time step per iteration, as masking more steps did not improve performance, and we apply our proposed masked event dropout with probability 0.5. Appendix A provides more details on hyperparameters. To show that our model generalises well without extensive tuning, we use the same hyperparameters for all tasks and datasets.

During fine-tuning, we find that additional regularization is needed given the sparsity of positive labels. We increase the Transformer dropout probability to 0.5, mask input time bins with probability 0.2, increase the weight decay to 0.01, and increase the warmup steps to 5000. Note that we continue to use input bin masking here as it provides effective regularization analogous to input dropout. We stop fine-tuning when the PR-AUC does not increase for 40 epochs on the validation set and take the model from the epoch with highest PR-AUC on the validation set.

**Baselines:** We show that TESS is competitive across a range of baseline models, including the well established XGBoost and LSTM baselines as well as state-of-the-art deep learning models:

- **XGBoost** (Chen & Guestrin, 2016): A scalable tree-based gradient boosting model that has been shown to outperform deep learning models on tabular data (Shwartz-Ziv & Armon, 2022).

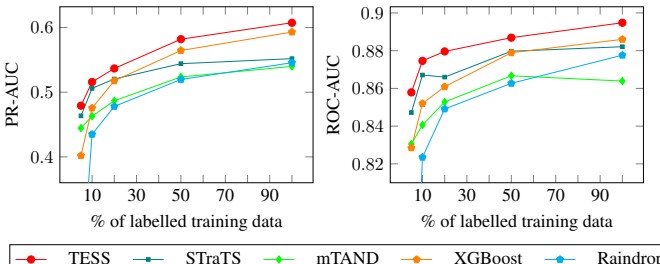

| Model | ROC-AUC | PR-AUC |
|-------|---------|--------|
| mTAND | $0.847 \pm 0.006$ | $0.473 \pm 0.016$ |
| STraTS | $0.875 \pm 0.001$ | $0.525 \pm 0.001$ |
| TESS | $\mathbf{0.8818 \pm 0.0004}$ | $\mathbf{0.5441 \pm 0.002}$ |

Table 2: Performance on the MIMIC-IV mortality prediction task with pre-trained encoders where the encoder weights are frozen during fine-tuning.

Figure 2: Performance on MIMIC-IV mortality prediction task using different percentages of labelled data for selected models.

- **LSTM** (Graves, 2012): A standard time series RNN. We use the same binned input format as for TESS.
- **mTAND** Shukla & Marlin (2021): An encoder-decoder based model that uses an attention module to interpolate irregular and sparse multivariate time series. Uses an unsupervised training task.
- **STraTS** (Tipirneni & Reddy, 2022): A Transformer-based model where every observation is embedded separately to produce the Transformer input sequence. Uses self-supervised pre-training.
- **Raindrop** Zhang et al. (2022): A graph-based neural network model that uses message passing between time series variables to learn relevant relationships.

We also compare to TESS without SSL, labelled TE(SS-). Results are shown in Table 1 and the details on baseline implementations are given in Appendix B. We report the mean and standard deviation of ROC-AUC and PR-AUC over five supervised training runs using different random seeds. We note that while ROC-AUC and PR-AUC demonstrate similar trends, PR-AUC provides more discrimination between methods.

**Results:** From Table 1 we see that TESS outperforms all baseline models on all datasets, except for XGBoost on MIMIC-IV pheotyping, where the performance of both models is on par. Notably, our tuned XGBoost model is the strongest baseline across both datasets. This indicates that XGBoost with appropriate feature engineering and hyperparameter tuning is still competitive with neural network models for sparse irregular time series, and should be included in evaluations of future methods. Compared to the Transformer-based baseline model, STraTS, TESS shows large consistent gains across tasks, with a $10\%$ increase in PR-AUC on MIMIC-IV mortality. These results suggest that the increased input granularity of STraTS relative to TESS is not required for these tasks. The performance of TE(SS-) shows that our SSL approach drives gains on both datasets. We investigate the impact of input resolution and SSL further in our ablation study.

**Representation Quality:** To evaluate the quality of the representations learned by our model compared to other baselines, we first carry out self-supervised pre-training, then freeze the encoder weights and fine-tune the model with a linear classifier attached to the encoder. Among our baselines, mTAND and STraTS can also be trained in this way. The original mTAND model augments supervised training with an unsupervised component, so for a fair comparison, we pretrain the mTAND encoder-decoder architecture using only the unsupervised loss. As shown in Table 2, TESS outperforms both of these baselines. This demonstrates the ability of TESS to learn useful patient representations from our self-supervised pre-training approach, without relying on labelled data. We also see that results for all models are lower than SSL combined with end-to-end fine-tuning in Table 1, indicating that it is preferable to fine-tune all weights.

We compare the performance of TESS with the baseline models when only a fraction of labelled data is used for supervised finetuning. Figure 2 shows that TESS outperforms the baselines consistently across all fractions of labelled data. The performance gap relative to the self-supervised Transformer baseline, STraTS, widens as more labelled data is available, while the gain of TESS over the XGBoost increases as the percentage of labelled data decreases, demonstrating the effectiveness of SSL in the sparse setting.

Figure 3 visualizes 768 dimensional representations learned by TESS on MIMIC-IV using t-SNE (Van der Maaten & Hinton, 2008). We first reduce the embedding dimension to 50 using PCA and classify each point

| Experiment | PR-AUC |
|---|---|
| **Baseline** | $0.607 \pm 0.003$ |
| No SSL | $0.552 \pm 0.007$ |
| SSL without value prediction | $0.554 \pm 0.004$ |
| SSL without mask prediction | $0.597 \pm 0.004$ |
| SSL without masked event dropout | $0.599 \pm 0.005$ |
| Omitting mask $m$ from inputs | $0.580 \pm 0.006$ |
| Using positional encoding for time | $0.589 \pm 0.008$ |
| No static input | $0.575 \pm 0.005$ |
| Binning with max aggregation | $0.607 \pm 0.003$ |
| Binning with mean aggregation | $0.604 \pm 0.007$ |

Table 3: Ablation study results measured on the MIMIC-IV mortality task.

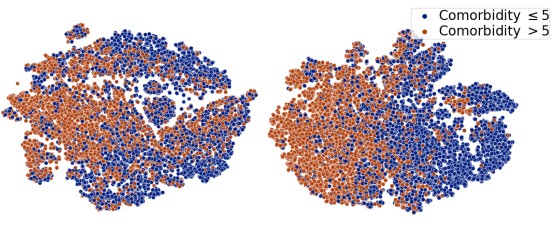

(a) With SSL pre-training    (b) With SSL pre-training and fine-tuning on Phenotyping

Figure 3: t-SNE visualization of the learned representations for the MIMIC-IV dataset. Each dot represents a patient categorized by the number of comorbities in their phenotyping target.

based on the number of comorbidities, defined as the sum of diagnoses in the phenotyping target. The figure demonstrates that TESS distinguishes cases with high comorbidities from those with low comorbidities after SSL pre-training and the separation improves with fine-tuning.

**Ablation Study:** To evaluate the importance of key components of our approach, including SSL and the input representation, we conduct an ablation study with results in Table 3. As noted from the results with TE(SS-) in Table 1, there is a large decrease in performance without self-supervised pre-training. We see from Table 3, that removing each component of our SSL approach, including value prediction, mask prediction and masked event dropout, leads to a considerable drop in performance. For the input representation, removing missingness information contained in the mask $m$ results in a significant performance decrease. We

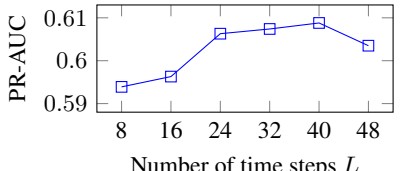

Figure 4: TESS PR-AUC on MIMIC-IV mortality using different numbers of time bins $L$.

see that using continuous value time embeddings is also useful, given the performance drop from switching to learned positional embeddings, which do not take into account the actual time values represented by each bin. We see that using different aggregation functions for binning such as taking a max or mean value makes no notable difference in the performance. Finally, omitting static variables from the Transformer input results in a significant drop in the performance as expected.

Figure 4 shows the effect on performance of varying $L$, the number of bins in the time series input. While the information loss from aggregating events in very few time bins is harmful, performance saturates above 40 time bins and starts to decrease. This shows that for EHR data, we can significantly compress the input time series without negatively impacting performance, likely due to nearby measurements being highly correlated. We hypothesize that the decrease in performance above $L = 40$ is caused by useful inputs being more sparse, which can be harmful for neural network models handling tabular data Grinsztajn et al. (2022).

## 5 CONCLUSION

We introduce TESS, a Transformer architecture for deep learning on sparse irregular time series data in health care. Our experiments show that this architecture is especially effective in learning useful information during self-supervised pre-training, allowing it to outperform state-of-the-art models. We believe that advancing the state of the art in self-supervised learning will help drive substantial improvements in future health care modelling. It is straightforward to extend TESS to multi-modal inputs, so future work can take advantage of a wider range of input data. We also expect that its approach is general enough to be applied to non-ICU EHR data and to sparse irregular time series data in other domains beyond health care.

## 6 REPRODUCIBILITY STATEMENT

We provide the details of our model architecture and hyperparameters necessary to replicate our model in Section 4 and Appendix A. Further, to allow better reproducibility, we run the experiments independently in Tables 1 to 3 and Figure 4 by presetting the random seeds to 5 different values from $2020 - 2025$. All other experiments were performed with random seed value set to 2020. We also provide the details of baseline implementations in Appendix B. We will be publishing implementation code for our model as well as the IDs for the patient-level splits on our lab's Github page along with the camera-ready version of the paper.

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

Table 4: XGBoost Hyperparameter Tuning Distributions

| Hyperparameter | Distribution |
|---|---|
| Number of rounds | Uniform on $\{50, 51, \ldots, 250\}$ |
| max_depth | Uniform on $\{2, 3, \ldots, 16\}$ |
| eta | Log-uniform on $[0.001, 1]$ |
| lambda | Log-uniform on $[0.001, 1]$ |
| alpha | Log-uniform on $[0.001, 1]$ |
| subsample | Uniform on $[0.2, 1]$ |
| min_child_weight | Log-uniform on $[0.01, 100]$ |

## A  ADDITIONAL TESS HYPERPARAMETERS

In our experiments, we use the following dimensions for the subnetworks within TESS. The static encoder is a single linear layer. The time series encoder has two hidden layers of size 256. The Transformer encoder has 4 layers, uses 4 heads in its multi-head attention blocks, and uses hidden layers of size 4096 in its feedforward networks. The self-supervised prediction heads have two hidden layers of size 256. The classification head has three hidden layers of size 128.

For the t-SNE plot, we used the implementation provided by scikit-learn[1] with the following hyperparameters: perplexity = 20, n_iter = 1000, learning_rate = 900, random_state = 2021, init = "pca".

## B  BASELINE DETAILS

For XGBoost, we use the same aggregated input representation as for TESS, with all $x$, $m$, and $s$ values concatenated into a feature vector. However, we find that XGBoost does not handle the sparsity of inputs well, and so we impute missing $x$ values using the last previously observed value when available. We perform random tuning with 100 tests using the hyperparameter distributions given in Table 4 and use the configuration with best PR-AUC on the validation set.

For mTAND, we use the configuration/hyperparameters given in Shukla & Marlin (2021) and their published code repository, using their PhysioNet hyperparameters for our PhysioNet tests and their MIMIC-III hyperparameters for our MIMIC-IV tests. Unlike the datasets evaluated in their paper, our MIMIC-IV phenotyping task uses arbitrarily long patient stays as input data, making it infeasible to train with the provided configurations. To mitigate this issue, for phenotyping only, we increase the quantization windows from 5 to 30 minutes and we limit the length of input data to the first two weeks of the patient stay. We also find that our zero-mean unit-variance normalization massively increases the mTAND reconstruction loss and reduces performance. For all tasks, we instead scale all variables to range from 0 to 1, matching their provided code. Further, we encode the static variables as time series with one sample as an input, which matches the mTAN code repository.

For Raindrop, we use the configuration/hyperparameters given in Zhang et al. (2022) and their published code repository for PhysioNet-2012. For our PhysioNet tests, we use the raw time steps given in the dataset and do not discretize time. Unlike PhysioNet, the set of time steps at which observations can be made in MIMIC-IV is not limited, making it infeasible to use raw inputs. To provide a fair comparison on MIMIC-IV, we use the same discretized time bins as for TESS. The static data is passed directly into the Raindrop model as their implementation also handles the static data along with the time series data.

---

[1]https://scikit-learn.org/stable/modules/generated/sklearn.manifold.TSNE.html

For STraTS, the configuration/hyperparameters are set according Tipirneni & Reddy (2022) and their published code repository. As suggested in the paper, we set the maximum number of observations to the $99^{th}$ percentile of the observations in the $48h$ observation window. This results in 1832 and 1898 maximum sequence length for MIMIC-IV and PhysioNet respectively. The static data for STraTS is passed through a feed-forward neural network to obtain the embedding before concatenating with the time series embedding and passing through the final dense layer as described in the original paper.

## C   ADDITIONAL RESULTS

The extended version of Figure 2 along with the performance of Raindrop and all the models evaluated at additional data points is given in Figure 5. We see that the same trend noted in Section 4 holds for this extensive evaluation.

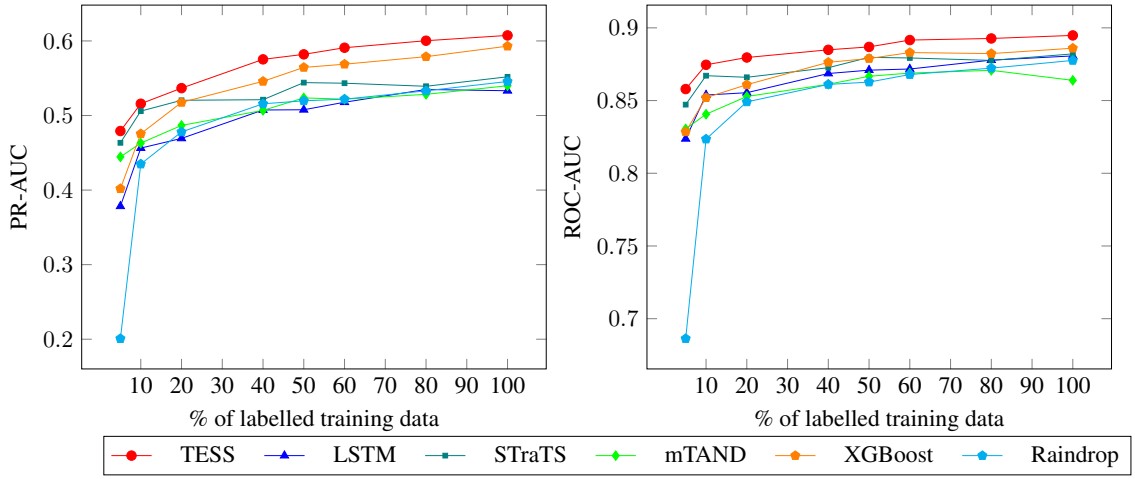

Figure 5: Performance on MIMIC-IV mortality prediction task using different percentages of labelled data.

To analyze the performance of TESS on different patient sub-groups, we evaluated our model on different groups of patients in MIMIC-IV dataset stratified according to various demographics such as age (Table 5), race (Table 7) and gender (Table 6). We see from Table 5 that the model performance is higher for the lower age group. This could be explained by the fact that the mortality risk varies across different age groups and it might be easier to predict mortality for some age groups compared to others. As shown in Table 6, TESS performs well across different gender groups. We also see that the number of samples for each sub-groups are fairly evenly distributed in both the training and test dataset. Table 7 shows the performance of TESSon different race groups. We see a low performance on the American Indian sub-group which can be explained by the very low number of samples in both the training and testing data. As the test dataset for American Indian race only consists of 13 samples, the performance metrics are not reliable.

| Age | ROC-AUC | PR-AUC | Test Samples | Train Samples |
|-----|---------|--------|--------------|---------------|
| 18-29 | 0.963 | 0.725 | 202 (3.6%) | 715 (3.63%) |
| 30-49 | 0.92 | 0.658 | 655 (11.8%) | 2608 (13.24%) |
| 50-59 | 0.911 | 0.661 | 951 (17.2%) | 3279 (16.65%) |
| 60-69 | 0.9 | 0.599 | 1346 (24.3%) | 4583 (23.27%) |
| 70 above | 0.864 | 0.587 | 2385 (43.1%) | 8514 (43.22%) |

Table 5: Performance of TESS on the MIMIC-IV mortality task on different sub-groups stratified by age along with the number of samples in train and test dataset.

| Gender | ROC-AUC | PR-AUC | Test Samples | Train Samples |
|--------|---------|--------|--------------|---------------|
| M | 0.882 | 0.608 | 3140 (56.7%) | 10964 (55.66%) |
| F | 0.902 | 0.597 | 2399 (43.3%) | 8735 (44.34%) |

Table 6: Performance of TESS on the MIMIC-IV mortality task on different sub-groups stratified by gender along with the number of samples in train and test dataset.

| Race | ROC-AUC | PR-AUC | Test Samples | Train Samples |
|------|---------|--------|--------------|---------------|
| Asian | 0.844 | 0.55 | 144 (2.6%) | 594 (3.02%) |
| White | 0.887 | 0.574 | 3729 (67.3%) | 13068 (66.34%) |
| American Indian | 0* | 0.125* | 13 (0.23%) | 39 (0.20%) |
| Black/African American | 0.907 | 0.545 | 578 (10.4%) | 2096 (10.64%) |
| Hispanic | 0.891 | 0.601 | 216 (3.9%) | 699 (3.55%) |
| Other | 0.834 | 0.627 | 236 (4.26%) | 809 (4.11%) |
| Missing | 0.8997 | 0.575 | 68 (1.23%) | 345 (1.75%) |
| Unknown | 0.898 | 0.723 | 555 (10.01%) | 2049 (10.40%) |

Table 7: Performance of TESS on the MIMIC-IV mortality task on different sub-groups stratified by race along with the number of samples in train and test dataset. *Metrics showing unrealistic results due to few test samples.

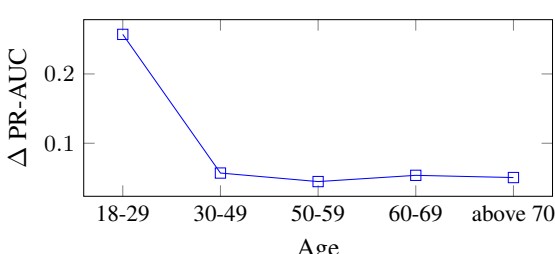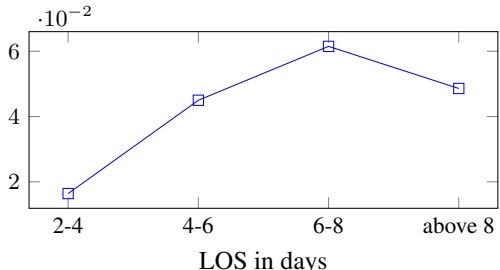

Figure 6: Performance gain with SSL pre-training task on the MIMIC-IV mortality prediction task. The patients are stratified into different sub-categories based on age and the length of the ICU stay (LOS).

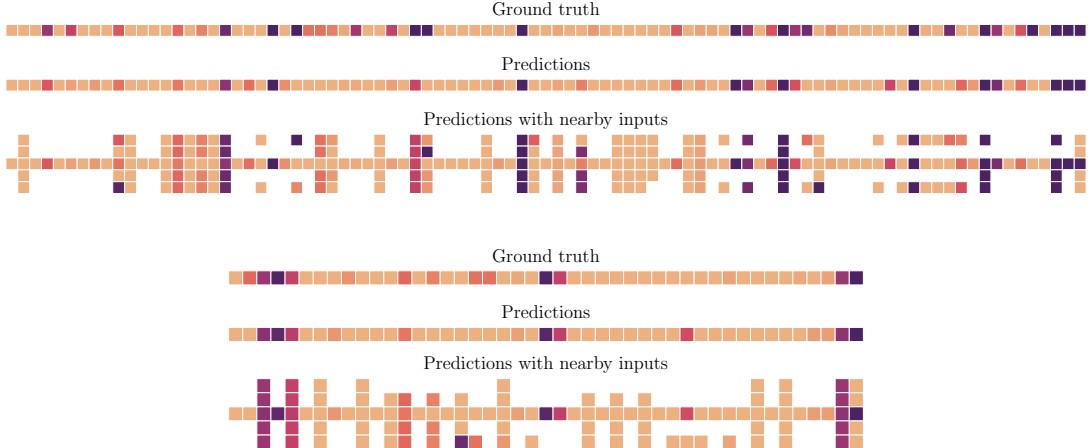

Figure 7: Sample ground truths and model predictions from masked value pre-training. Only variables that are present in the masked time step are shown. The bottom plot shows the model output in the context of nearby input time steps to show when the model can and cannot interpolate from nearby data.

Figure 6 shows the performance gain across different sub-groups with SSL pre-training. We stratify MIMIC-IV mortality test dataset based on different metadata such as age of the patient, and patient's length of stay (LOS) in the ICU and evaluate TESS and TE(SS-) on each sub-category. We see a gain from SSL pre-training across all the age and LOS strata. We can also see that the gain in the performance increases with the LOS which indicates that the pre-training is able to learn a good representation across time.

Figure 7 visualizes some sample outputs from the value prediction head after pre-training, comparing them to ground truth values and showing them in the context of nearby time steps that are not masked. The effects of our proposed event type dropout can be seen here, as many events that the model could otherwise use to predict inputs using interpolation have been removed entirely. Despite this added difficulty, the model can still make good predictions for the dropped event types.

# D  MIMIC-IV BENCHMARK DETAILS

We use all variables from Harutyunyan et al. (2019) except GCS total, which does not have a corresponding item ID in MIMIC-IV, plus chart and lab variables observed in 50% or more of patient stays. This amounts to 85 chart event variables and 29 lab event variables. Similarly, we include 9 different static variables. All variables are given in Table 8. The categorical variables are encoded using one-hot encoding giving us a 47-dimensional vector. Our training set consists of a total of 19,699 instances with a positive mortality rate of 12.95%, our validation set contains 4,257 instances with a mortality rate of 13.55%, and our test set contains 4,245 instances with a mortality rate of 12.39%.

| Variable | Type | Source | Item IDs |
|---|---|---|---|
| **Variables from Harutyunyan et al. (2019)** | | | |
| Capillary refill rate | Time series | | 223951, 224308 |
| Diastolic blood pressure | Time series | | 220051, 220180, 224643, 225310, 227242 |
| Fraction inspired oxygen | Time series | | 223835 |
| Glasgow coma scale eye opening | Time series | | 220739 |
| Glasgow coma scale verbal response | Time series | | 223900 |
| Glasgow coma scale motor response | Time series | | 223901 |
| Glucose | Time series | | 220621, 225664, 226537, 228388 |
| Heart rate | Time series | | 220045 |
| Height | Time series | | 226707, 226730 |
| Mean blood pressure | Time series | | 220052, 220181 |
| Oxygen saturation | Time series | | 220227, 220277 |
| Respiratory rate | Time series | | 220210, 223851, 224689, 224690 |
| Systolic blood pressure | Time series | | 220050, 220179, 224167, 225309, 227243 |
| Temperature | Time series | | 223761, 223762, 224027 |
| Weight | Time series | | 224639, 226512, 226531 |
| pH | Time series | | 220274, 220734, 223830, 228243 |
| **Additional time series variables** | | | |
| Heart Rate | Time series | ICU Chartevents | 220045 |
| O2 saturation pulseoxymetry | Time series | ICU Chartevents | 220277 |
| Respiratory Rate | Time series | ICU Chartevents | 220210 |
| GCS - Eye Opening | Time series | ICU Chartevents | 220739 |
| GCS - Verbal Response | Time series | ICU Chartevents | 223900 |
| GCS - Motor Response | Time series | ICU Chartevents | 223901 |
| Alarms On | Time series | ICU Chartevents | 224641 |
| Parameters Checked | Time series | ICU Chartevents | 224168 |
| Heart Rate Alarm - Low | Time series | ICU Chartevents | 220047 |
| Heart rate Alarm - High | Time series | ICU Chartevents | 220046 |
| Non Invasive Blood Pressure mean | Time series | ICU Chartevents | 220181 |
| Non Invasive Blood Pressure systolic | Time series | ICU Chartevents | 220179 |
| Non Invasive Blood Pressure diastolic | Time series | ICU Chartevents | 220180 |
| O2 Saturation Pulseoxymetry Alarm - Low | Time series | ICU Chartevents | 223770 |

| | | | |
|---|---|---|---|
| O2 Saturation Pulseoxymetry Alarm - High | Time series | ICU Chartevents | 223769 |
| Resp Alarm - High | Time series | ICU Chartevents | 224161 |
| Resp Alarm - Low | Time series | ICU Chartevents | 224162 |
| Braden Sensory Perception | Time series | ICU Chartevents | 224054 |
| Braden Mobility | Time series | ICU Chartevents | 224057 |
| Braden Moisture | Time series | ICU Chartevents | 224055 |
| Braden Activity | Time series | ICU Chartevents | 224056 |
| Braden Nutrition | Time series | ICU Chartevents | 224058 |
| Braden Friction/Shear | Time series | ICU Chartevents | 224059 |
| SpO2 Desat Limit | Time series | ICU Chartevents | 226253 |
| Temperature Fahrenheit | Time series | ICU Chartevents | 223761 |
| IV/Saline lock | Time series | ICU Chartevents | 227344 |
| Gait/Transferring | Time series | ICU Chartevents | 227345 |
| Ambulatory aid | Time series | ICU Chartevents | 227343 |
| Mental status | Time series | ICU Chartevents | 227346 |
| Secondary diagnosis | Time series | ICU Chartevents | 227342 |
| History of falling (within 3 mnths) | Time series | ICU Chartevents | 227341 |
| Potassium (serum) | Time series | ICU Chartevents | 227442 |
| Sodium (serum) | Time series | ICU Chartevents | 220645 |
| Chloride (serum) | Time series | ICU Chartevents | 220602 |
| Creatinine (serum) | Time series | ICU Chartevents | 220615 |
| BUN | Time series | ICU Chartevents | 225624 |
| HCO3 (serum) | Time series | ICU Chartevents | 227443 |
| Anion gap | Time series | ICU Chartevents | 227073 |
| Hematocrit (serum) | Time series | ICU Chartevents | 220545 |
| Glucose (serum) | Time series | ICU Chartevents | 220621 |
| Hemoglobin | Time series | ICU Chartevents | 220228 |
| Platelet Count | Time series | ICU Chartevents | 227457 |
| WBC | Time series | ICU Chartevents | 220546 |
| Magnesium | Time series | ICU Chartevents | 220635 |
| Non-Invasive Blood Pressure Alarm - Low | Time series | ICU Chartevents | 223752 |
| Non-Invasive Blood Pressure Alarm - High | Time series | ICU Chartevents | 223751 |
| Phosphorous | Time series | ICU Chartevents | 225677 |
| Calcium non-ionized | Time series | ICU Chartevents | 225625 |
| Pain Level | Time series | ICU Chartevents | 223791 |
| Richmond-RAS Scale | Time series | ICU Chartevents | 228096 |
| Prothrombin time | Time series | ICU Chartevents | 227465 |
| INR | Time series | ICU Chartevents | 227467 |
| PTT | Time series | ICU Chartevents | 227466 |
| Capillary Refill R | Time series | ICU Chartevents | 223951 |
| Capillary Refill L | Time series | ICU Chartevents | 224308 |
| Admission Weight (lbs.) | Time series | ICU Chartevents | 226531 |
| Goal Richmond-RAS Scale | Time series | ICU Chartevents | 228299 |
| ST Segment Monitoring On | Time series | ICU Chartevents | 228305 |
| O2 Flow | Time series | ICU Chartevents | 223834 |
| Glucose finger stick (range 70-100) | Time series | ICU Chartevents | 225664 |
| Pain Level Response | Time series | ICU Chartevents | 224409 |
| Intravenous / IV access prior to admission | Time series | ICU Chartevents | 225103 |
| 20 Gauge Dressing Occlusive | Time series | ICU Chartevents | 227368 |
| Strength R Arm | Time series | ICU Chartevents | 228412 |
| Strength L Arm | Time series | ICU Chartevents | 228409 |
| Strength R Leg | Time series | ICU Chartevents | 228411 |
| Strength L Leg | Time series | ICU Chartevents | 228410 |
| 20 Gauge placed in outside facility | Time series | ICU Chartevents | 226138 |
| Insulin pump | Time series | ICU Chartevents | 228236 |

| | | | |
|---|---|---|---|
| Self ADL | Time series | ICU Chartevents | 225092 |
| 20 Gauge placed in the field | Time series | ICU Chartevents | 228100 |
| History of slips / falls | Time series | ICU Chartevents | 225094 |
| High risk (¿51) interventions | Time series | ICU Chartevents | 227349 |
| Lactic Acid | Time series | ICU Chartevents | 225668 |
| Home TF | Time series | ICU Chartevents | 228648 |
| ETOH | Time series | ICU Chartevents | 225106 |
| Pressure Ulcer Present | Time series | ICU Chartevents | 228649 |
| Difficulty swallowing | Time series | ICU Chartevents | 225118 |
| 18 Gauge Dressing Occlusive | Time series | ICU Chartevents | 227367 |
| 18 Gauge placed in outside facility | Time series | ICU Chartevents | 226137 |
| Eye Care | Time series | ICU Chartevents | 225184 |
| Visual / hearing deficit | Time series | ICU Chartevents | 225087 |
| Currently experiencing pain | Time series | ICU Chartevents | 225113 |
| Dialysis patient | Time series | ICU Chartevents | 225126 |
| Daily Weight | Time series | ICU Chartevents | 224639 |
| Potassium | Time series | ICU Labevents | 50971 |
| Chloride | Time series | ICU Labevents | 50902 |
| Sodium | Time series | ICU Labevents | 50983 |
| Creatinine | Time series | ICU Labevents | 50912 |
| Urea Nitrogen | Time series | ICU Labevents | 51006 |
| Bicarbonate | Time series | ICU Labevents | 50882 |
| Anion Gap | Time series | ICU Labevents | 50868 |
| Glucose | Time series | ICU Labevents | 50931 |
| Hematocrit | Time series | ICU Labevents | 51221 |
| Platelet Count | Time series | ICU Labevents | 51265 |
| White Blood Cells | Time series | ICU Labevents | 51301 |
| Hemoglobin | Time series | ICU Labevents | 51222 |
| Red Blood Cells | Time series | ICU Labevents | 51279 |
| MCV | Time series | ICU Labevents | 51250 |
| MCH | Time series | ICU Labevents | 51248 |
| MCHC | Time series | ICU Labevents | 51249 |
| RDW | Time series | ICU Labevents | 51277 |
| Magnesium | Time series | ICU Labevents | 50960 |
| Phosphate | Time series | ICU Labevents | 50970 |
| Calcium, Total | Time series | ICU Labevents | 50893 |
| PT | Time series | ICU Labevents | 51274 |
| INR(PT) | Time series | ICU Labevents | 51237 |
| PTT | Time series | ICU Labevents | 51275 |
| pH | Time series | ICU Labevents | 50820 |
| Lactate | Time series | ICU Labevents | 50813 |
| Base Excess | Time series | ICU Labevents | 50802 |
| pO2 | Time series | ICU Labevents | 50821 |
| pCO2 | Time series | ICU Labevents | 50818 |
| Calculated Total CO2 | Time series | ICU Labevents | 50804 |
| **Static tabular variables** | | | |
| Age | Binary Variable | Admission Table | |
| Gender | Binary Variable | Admission Table | |
| English Language | Binary Variable | Admission Table | |
| Marital Status | Categorical Variable | Admission Table | |
| Insurance | Categorical Variable | Admission Table | |
| Admission Location | Categorical Variable | Admission Table | |
| Admission Type | Categorical Variable | Admission Table | |
| Race | Categorical Variable | Admission Table | |

| First Care Unit | Categorical Variable | ICU Admission Table |
| Observation Window Length | Continuous Variable | Derived |

Table 8: Time series and static variables used from MIMIC-IV dataset.

Following Harutyunyan et al. (2019), the phenotyping task has 25 binary target variables, corresponding to whether the following conditions were billed during the stay:

- Acute and unspecified renal failure
- Acute cerebrovascular disease
- Acute myocardial infarction
- Cardiac dysrhythmias
- Chronic kidney disease
- Chronic obstructive pulmonary disease and bronchiectasis
- Complications of surgical procedures or medical care
- Conduction disorders
- Congestive heart failure; nonhypertensive
- Coronary atherosclerosis and other heart disease
- Diabetes mellitus with complications
- Diabetes mellitus without complication
- Disorders of lipid metabolism
- Essential hypertension
- Fluid and electrolyte disorders
- Gastrointestinal hemorrhage
- Hypertension with complications and secondary hypertension
- Other liver diseases
- Other lower respiratory disease
- Other upper respiratory disease
- Pleurisy; pneumothorax; pulmonary collapse
- Pneumonia (except that caused by tuberculosis or sexually transmitted disease)
- Respiratory failure; insufficiency; arrest (adult)
- Septicemia (except in labor)
- Shock

This task has a total of $54,024$ training instances, $11,401$ validation instances and $11,509$ testing instances.

