# OpenReview forum: "Effective Self-Supervised Transformers For Sparse Time Series Data"
_ICLR.cc/2023/Conference — Submitted to ICLR 2023_

### Official Review · Reviewer_EBR4 · 2022-10-24

**Confidence:** 3
**Correctness:** 3
**Technical Novelty And Significance:** 3
**Empirical Novelty And Significance:** 2
**Recommendation:** 6

**Clarity, Quality, Novelty And Reproducibility:**

In general, the clarity and the quality of this work are good with several contributions to EHR data processing. The novelty is a bit limited as mentioned above in the Weaknesses section. It seems to be able to reproduce the proposed method, but some technical details could have been better presented with sufficient details for better reproducibility.

**Strength And Weaknesses:**

**Strengths**

\+ This paper proposed a feasible and effective approach for self-supervised learning on EHR data.

\+ Each of the designed/used components within the proposed architecture was well-motivated.

\+ The proposed method was shown to perform better than the compared previous methods.

\+ An ablation study was presented to show the effectiveness of the components within the proposed method.

\+ The paper is generally well-written and easy to follow.


**Weaknesses**

\- The technical novelty and contributions are a bit limited. The overall idea of using a transformer to process time series data is not new, as also acknowledged by the authors. The masked prediction was also used in prior works e.g. MAE (He et al., 2022). The main contribution, in this case, is the data pre-processing approach that was based on the bins. The continuous value embedding (CVE) was also from a prior work (Tipirneni & Reddy 2022), and also the early fusion instead of late fusion (Tipirneni & Reddy, 2022; Zhang et al., 2022). It would be better to clearly clarify the key novelty compared to previous works, especially the contribution (or performance gain) from the data pre-processing scheme.

\- It is unclear if there are masks applied to all the bins, or only to one bin as shown in Fig. 1.

\- It is unclear how the static data (age, gender etc.) were encoded to input to the MLP. The time-series data was also not clearly presented.

\- It is unclear what is the "learned [MASK] embedding" mean in the SSL pre-training stage of the proposed method.

\- The proposed "masked event dropout scheme" was not clearly presented. Was this dropout applied to the ground truth or the prediction? If it was applied to the prediction or the training input data, will this be considered for the loss function?

\- The proposed method was only evaluated on EHR data but claimed to be a method designed for "time series data" as in both the title and throughout the paper. Suggest either tone-down the claim or providing justification on more other time series data.

\- The experimental comparison with other methods seems to be a bit unfair. As the proposed method was pre-trained before the fine-tuning stage, it is unclear if the compared methods were also initialised with the same (or similar scale) pre-trained model. If not, as shown in Table 1, the proposed method without SSL performs inferior to most of the compared methods.

\- Missing reference to the two used EHR datasets at the beginning of Sec. 4.

**Summary Of The Paper:**

This paper presented a new method for self-supervised learning on sparse time series data. Specifically, Transformers for EHR data with self-supervised learning -- TESS was proposed with an input binning scheme and a combination of both missingness masks and event values. Experimental analysis on two public EHR datasets shows the effectiveness of the proposed method. The main contributions of this paper are the proposed bin-based data pre-processing scheme and the joint processing model for multiple data types in EHR applications.

**Summary Of The Review:**

This paper is generally okay with clear motivation and a new approach to process EHR data with shown better performance on two widely used public datasets. There are a few concerns regarding the novelty and contributions, experiments and some statements/claims, but they could possibly be addressed within the rebuttal phase. As a result, I recommend a positive rating at the current stage and may change my recommendation after the rebuttal.

---

> ### Author Response · Authors · 2022-11-17
> **Response to reviewer EBR4**
>
> Thank you for the useful suggestions and for the positive rating. We hope we've addressed your concerns about novelty and reproducibility in our common response, and we've also made some paper edits and provided some clarifications in response to the other matters you raised below.
>
>
> > It is unclear if there are masks applied to all the bins, or only to one bin as shown in Fig. 1.
>
> Masks can generally be applied to any subset of bins, and the fraction of bins to randomly mask can be treated as a hyperparameter. For our experiments, we apply the mask to only one random bin per instance in the batch. We've edited the description of masking to clarify that this is an option.
>
> > It is unclear how the static data (age, gender etc.) were encoded to input to the MLP.
>
> We use different encoding techniques depending on the type of the static data. For numeric and binary variables such as age and gender, we use the original value. For categorical variables such as marital status, race, etc., we use one-hot encoding as described in Appendix D. We generally consider this to be a task-specific implementation detail, as different encoding schemes could be used with our model.
>
>
> > It is unclear what is the "learned [MASK] embedding" mean in the SSL pre-training stage of the proposed method.
>
> Learned embeddings are randomly initialized vectors of weights that are learned during training, making them specific to the training task.
>
> > The proposed "masked event dropout scheme" was not clearly presented. Was this dropout applied to the ground truth or the prediction? If it was applied to the prediction or the training input data, will this be considered for the loss function?
>
> Thank you for raising this, we've edited the description in the paper to clarify. The masked event dropout scheme is applied to the time series input during the SSL pre-training stage to prevent the model from learning to reconstruct events by simply interpolating from nearby time bins. As such, for the events observed in the masked time bins i.e., with some $m_{l,i}=1$, a fraction of them have *all* observed events of the same type dropped, by setting their value to 0 across all other time bins. The loss function is only applied to the prediction of the masked time bin, hence, the loss function does not consider the masked event dropout scheme directly.
>
> > The proposed method was only evaluated on EHR data but claimed to be a method designed for "time series data" as in both the title and throughout the paper. Suggest either tone-down the claim or providing justification on more other time series data.
>
> Thank you for raising this, we've edited the abstract and body of the paper to refine this claim to focus on ICU EHR data. While we only validate the model on EHR data, the design of the model is not specific to EHR data in any way, and it could be applied to other domains in future work (a lack of large sparse irregular time series datasets means that most papers in this area focus on EHR data).
>
> > As the proposed method was pre-trained before the fine-tuning stage, it is unclear if the compared methods were also initialised with the same (or similar scale) pre-trained model. If not, as shown in Table 1, the proposed method without SSL performs inferior to most of the compared methods.
>
> This is an important point, so thank you for raising it - we've edited the baseline descriptions to clarify this. mTAND and STraTS also use a self-supervised or unsupervised training procedure, so they provide a comparison with existing models that use a similar procedure to ours. STraTS is pretrained as outlined in [1] followed by fine-tuning for each downstream task. Similarly, mTAND uses both unsupervised and supervised training tasks as outlined in [2]. In general, we don't intend TE(SS-) to show competitive performance, since SSL is a key compoment of our proposed method.
>
> > Missing reference to the two used EHR datasets at the beginning of Sec. 4.
>
> Thank you for pointing this out. We have added the corresponding references in the paper.
>
> [1] Sindhu Tipirneni and Chandan K. Reddy. Self-supervised transformer for sparse and irregularly sampled multivariate clinical time-series. ACM Transactions on Knowledge Discovery from Data, 1(1), 2022.
> [2] Satya Narayan Shukla and Benjamin Marlin. Multi-time attention networks for irregularly sampled time series. In International Conference on Learning Representations, 2021

---

### Official Review · Reviewer_HAwr · 2022-10-25

**Confidence:** 5
**Correctness:** 2
**Technical Novelty And Significance:** 3
**Empirical Novelty And Significance:** 2
**Recommendation:** 5

**Clarity, Quality, Novelty And Reproducibility:**

The presentation of the paper lends itself well for comprehension and the details provided, along with the datasets used, should make the paper verifiable. The proposed SSL method is novel and can be important by its own merit


**Details Of Ethics Concerns:**

The authors haven't conducted any group level analysis. If the model is to be applied in a real-world setting this is of paramount importance


**Strength And Weaknesses:**

Some of the key strengths of the paper are as follows:

- The paper studies a very important problem that is prevalent in longitudinal EHR data. The results against the chosen baselines show strong performance and may have significant impact in using AI for clinical settings. Of special note the method performs better than XGBoost (see more below).
- The proposed method of self-supervision is novel and simple. While the method is proposed for transformer based architecture, it could be adopted to most architectures such as LSTM as well other forms of attention based models.
- The paper is well presented and the illustrations makes the paper easy to follow. The variables used in the model have been communicated. The experiments have also been conducted on two well known benchmarks and the results should be reproducible with some level of effort.

The paper can be improved upon by addressing some of the concerns below:

- While the identified problem is well motivated, the clinical significance seems to lack sufficient depth. Compared to other forms of EHR, ICU data is well populated with less randomly missing data points. Also, typical EHR records, especially for chronic diseases, can go back for decades with large periods of data that are irrelevant to prediction for the time point of interest. The claim of the paper as a general solution for EHR data is thus not well supported.
- Regarding novelty, the first aspect of the proposed method considers aggregation of data into bins, while novel in a model integrated manner, is a well known and often used pre-processing technique to handle EHR data. In fact, other forms of aggregation such as those based on higher level EHR concepts have also been used to reduce data sparsity.
- In order to interpret the usefulness of the proposed method, it is critical to understand the performance over different sub-groups. At the very minimum, a typical "Table one" describing the data distribution should be presented. It is also important to understand whether the self-supervision can amplify biases in the data and discuss any proposed solutions to that


**Summary Of The Paper:**

The paper presents a novel solution to effectively handle the sparsity of data in EHR using a combination of techniques including self-supervision and aggregation based normalization of sparse inputs. The authors compared their model on two real world datasets for three different tasks. They claim that the scheme effectively improves modeling efficacy over state of the art methods


**Summary Of The Review:**

The authors have analyzed an important and well-known problem in clinical AI.
While the experiments on the chosen baseline is promising, the claims, especially considering the clinical perspective, is not well substantiated. Of note, the SSL method by itself is novel and from the results seem to contribute the most to the model performance. The method can be arguably applied to other architectures as well. It will be useful for the authors to deliberate on the claims e.g. either narrow the claims to ICU (i.e more regular) settings or provide experimental validations on longer term EHR data. Apart from the aforementioned aspects the authors may also want to consider the following aspects:

- Provide intuitions in the paper (rather than form citation) about the CVE approach to capture temporal information over simpler methods such as cosine embedding
- From Table 1, it seems TE(SS-) doesn't perform better than STraTS. This may indicate that SSL is the main driver of the performance. With this in mind, the authors may consider a simple extension of the SSL to base models such as LSTM or more classical DL method for clinical AI such as RETAIN and SAnD.
- In the implementation section, the authors note that
> We apply zero-mean and unit-standard deviation normalization of all inputs in x. We also clip outliers using a threshold of three median absolute deviations from the median. These steps allow for stable training without requiring clinical knowledge of normal variable ranges.
    * While this is indeed beneficial from a ML modeling standpoint, it is not always clinically reasonable to neglect domain information such as typical values of clinical variables. EHR data is often noisy and such steps are often crucial to ensure that the model learns from reasonable manifestations of patient data
- The authors claim that TESS handle the temporal granularity in a simpler manner than StraTS. While some justification is provided, it may be important to quantify these using some measures such as perplexity and/or computational requirement.
- The discussions are model interpretations are not well supported. While t-SNE plots have been reported in Figure 3 there are two important problems with the analysis; (a) t-SNE can provide widely different manifolds based on the hyper-parameters. These should be reported. Comparison with other stable methods such as PCA could be considered. (b) the plots in Figure 3b can be argued to have some overlap - it is not evident from the naked eye whether TESS produces well separated representations.
- On a similar note, for the model to be of real-world usage, the authors may want to investigate on the feature importance/attributions driving the improved performance

---

> ### Author Response · Authors · 2022-11-17
> **Response to reviewer HAwr (1/2)**
>
> Thank you for the useful feedback and clinical insights. We've edited the paper to refine our claims and presentation, and added some additional information as suggested.
>
> > While the identified problem is well motivated, the clinical significance seems to lack sufficient depth. Compared to other forms of EHR, ICU data is well populated with less randomly missing data points. Also, typical EHR records, especially for chronic diseases, can go back for decades with large periods of data that are irrelevant to prediction for the time point of interest. The claim of the paper as a general solution for EHR data is thus not well supported.
>
> Thank you for raising this, we've updated the abstract and body to clarify that our results and model are focused on ICU data, although we believe the  structure and data processing approach is general enough that the model could be used with longer term EHR data and other sparse irregular time series data.
>
> > Regarding novelty, the first aspect of the proposed method considers aggregation of data into bins, while novel in a model integrated manner, is a well known and often used pre-processing technique to handle EHR data. In fact, other forms of aggregation such as those based on higher level EHR concepts have also been used to reduce data sparsity.
>
> We agree that our use of binning is only novel with respect to its use in our model: to enable the use of large self-supervised transformers on EHR data. We've updated the abstract and body of the paper to clarify this claim. We do think it's valuable, though, that our proposed method can achieve good results without the explicit application of domain knowledge, likely due to our use of representation learning. The broader matter of novelty is addressed in our response to all reviewers.
>
> > In order to interpret the usefulness of the proposed method, it is critical to understand the performance over different sub-groups. At the very minimum, a typical "Table one" describing the data distribution should be presented. It is also important to understand whether the self-supervision can amplify biases in the data and discuss any proposed solutions to that.
>
> Thank you for this suggestion, we have added information on the data distribution and our performance across subgroups along with the discussion in Appendix C.
>
>
> > Apart from the aforementioned aspects the authors may also want to consider the following aspects:
> >
> >> Provide intuitions in the paper (rather than form citation) about the CVE approach to capture temporal information over simpler methods such as cosine embedding
>
> We have added some additional clarification on the role of CVE to the paper in Section 3.1. We would also like to highlight that our ablation study shows that CVE outperforms positional encoding, which supports its use in our method.
>
> > From Table 1, it seems TE(SS-) doesn't perform better than STraTS. This may indicate that SSL is the main driver of the performance. With this in mind, the authors may consider a simple extension of the SSL to base models such as LSTM or more classical DL method for clinical AI such as RETAIN and SAnD.
>
> We agree that SSL drives much of our performance, but we also specifically design our model to benefit from SSL. STraTS is also a transformer model trained with SSL, but even with this pre-training, we significantly outperform it, showing the advantages of our design. We've updated the baseline descriptions to emphasize which models did and didn't use pre-training.
>
> > In the implementation section, the authors note that
> >
> >  _We apply zero-mean and unit-standard deviation normalization of all inputs in x. We also clip outliers using a threshold of three median absolute deviations from the median. These steps allow for stable training without requiring clinical knowledge of normal variable ranges._
> >
> > While this is indeed beneficial from a ML modeling standpoint, it is not always clinically reasonable to neglect domain information such as typical values of clinical variables. EHR data is often noisy and such steps are often crucial to ensure that the model learns from reasonable manifestations of patient data
>
> We agree that for practical ML deployments, using clinical knowledge to handle data can be important. However, our model's performance when dealing with real EHR data shows that it is robust without such clinical feature engineering, even when outliers and noise exist, which could significantly simplify practical applications.

---

> > ### Author Response · Authors · 2022-11-17
> > **Response to reviewer HAwr (2/2)**
> >
> > > The authors claim that TESS handle the temporal granularity in a simpler manner than StraTS. While some justification is provided, it may be important to quantify these using some measures such as perplexity and/or computational requirement.
> >
> > Thank you for this useful suggestion, we've edited the paper to note that our method has memory and computational complexity that is quadratic in the number of bins rather than the number of observed events, as in STraTS. Formally, our method is $O(L^2)$ rather than $O(n_p^2)$ where $L \ll n_p$ in general.
> >
> > > While t-SNE plots have been reported in Figure 3 there are two important problems with the analysis; (a) t-SNE can provide widely different manifolds based on the hyper-parameters. These should be reported. Comparison with other stable methods such as PCA could be considered. (b) the plots in Figure 3b can be argued to have some overlap - it is not evident from the naked eye whether TESS produces well separated representations.
> >
> > - For the t-SNE plots, we used the implementation provided by scikit-learn and have updated Appendix A of the paper to include the relevant hyperparameters we used.
> > - Additionally, we can quantitatively evaluate the quality of representations learnt by TESS from our experiment using a frozen encoder; please see Table 2 in the paper. This demonstrates that the  pre-trained representations are capturing useful patient information.
> >
> > > On a similar note, for the model to be of real-world usage, the authors may want to investigate on the feature importance/attributions driving the improved performance
> >
> > We thank the reviewer for the suggestion. Due to the limitations from space constraints in the paper, we leave explainability and other investigations of the model for future work.

---

### Official Review · Reviewer_fWW7 · 2022-10-25

**Confidence:** 5
**Clarity, Quality, Novelty And Reproducibility:** 1. The paper is readable, but the wri…
**Correctness:** 2
**Technical Novelty And Significance:** 2
**Empirical Novelty And Significance:** 2
**Recommendation:** 3

**Strength And Weaknesses:**

Strength:
1. The authors present a transformer model to represent the patients’ static variables and dynamic observations as fixed-size vectors.

Weakness:
1. The novelty is somewhat incremental.
    a) Model training: SSL pre-training and fine-tuning are also common tricks for deep learning models [1,2,8]. The work is an application of Transformer to the healthcare domain.
    b) Model framework: Lots of existing studies (e.g., [5,6,7]) have combined static variables and dynamic variables when learning patient representations and considering the time intervals and missing value problems.

2. The aggregation methods (e.g., mean, maximal, minimal, last observed variable) used in the paper might not be enough for real-world clinical settings. The frequency of lab tests could imply the patients’ disease severity. A simple aggregation might cause information loss.

3. The experiments could be improved:
    a) It would be better if the authors conducted experiments to show which aggregation methods are better for clinical settings (e.g., mean, maximal, minimal, last observed variable)?
    b) The lengths of different patients’ EHR data vary a lot (e.g., from 1~3 days to several weeks in ICUs). It is tricky to select the value of L for different patients. It would be better if the authors conducted experiments to show how the model is sensitive to L for different groups of patients, especially for patients with extremely short or long observation periods. Moreover, in Figure 4, when L is set 48, the performance becomes worse, which is unusual and worth explaining why.
    c) The statistics of dataset (e.g., age, gender) are missing.
    d) It is unclear how to learn patient representation [REP].

4. Experiment results show that the proposed model performs just as comparable to baselines on some tasks (e.g., phenotyping).

5. The paper writing could be improved. For example, the authors claim that “the use of Transformer in modelling sparse irregular time series with tabular data has not been widely explored.” However, Transformer has been widely used in modeling EHR data (e.g., [1-4]).

6. The implementation code is not available.

References:

[1] Rasmy L, Xiang Y, Xie Z, et al. Med-BERT: pretrained contextualized embeddings on large-scale structured electronic health records for disease prediction. NPJ digital medicine, 2021, 4(1): 1-13.

[2] Li Y, Rao S, Solares J R A, et al. BEHRT: transformer for electronic health records. Scientific reports, 2020, 10(1): 1-12.

[3] Li F, Jin Y, Liu W, et al. Fine-tuning bidirectional encoder representations from transformers (BERT)–based models on large-scale electronic health record notes: an empirical study. JMIR medical informatics, 2019, 7(3): e14830.

[4] Zhang X, Qian B, Cao S, et al. INPREM: an interpretable and trustworthy predictive model for healthcare//Proceedings of the 26th ACM SIGKDD International Conference on Knowledge Discovery & Data Mining. 2020: 450-460.

[5] Baytas I M, Xiao C, Zhang X, et al. Patient subtyping via time-aware LSTM networks//Proceedings of the 23rd ACM SIGKDD international conference on knowledge discovery and data mining. 2017: 65-74.

[6] Tipirneni S, Reddy C K. Self-supervised transformer for multivariate clinical time-series with missing values. arXiv preprint arXiv:2107.14293, 2021.

[7] Yuan Luo, Peter Szolovits, Anand Dighe, and Jason Baron. 2018. 3D-MICE: integration of cross-sectional and longitudinal imputation for multi-analyte longitudinal clinical data. JAMIA 25, 6 (2018), 645–653.

[8] Devlin J, Chang M W, Lee K, et al. Bert: Pre-training of deep bidirectional transformers for language understanding. arXiv preprint arXiv:1810.04805, 2018.


**Summary Of The Paper:**

The paper focuses on the robust representation of sparse time series EHR data. The authors propose a new model TESS, Transformers for EHR data with Self Supervised learning, a self-supervised Transformer-based architecture, to represent the patients’ data as fixed-size vectors with the consideration of irregular observations and missing values. The experiments on real-world datasets show that the proposed model outperforms the baselines.

**Summary Of The Review:**

The authors present a transformer model to represent the patients’ static variables and dynamic observations as fixed-size vectors. The paper is neither novel from machine learning nor from clinical informatics. There are also some major concerns in the experiments.

---

> ### Author Response · Authors · 2022-11-17
> **Response to reviewer fWW7**
>
> Thank you for the constructive feedback. We've discussed some overall concerns raised in our response to all reviewers. We've also provided additional clarifications and experiments in the paper, and we address individual concerns below:
>
> > The aggregation methods (e.g., mean, maximal, minimal, last observed variable) used in the paper might not be enough for real-world clinical settings. The frequency of lab tests could imply the patients’ disease severity. A simple aggregation might cause information loss.
>
> We agree that aggregation causes information loss, but our general empirical finding has been that aggregated inputs are sufficient for a variety of clinical prediction tasks (e.g., MIMIC-IV phenotyping includes performance across 25 different clinical targets). Using aggregation reduces the number of uninformative inputs, which tends to benefit deep learning models [1], and in the case of Transformers, using a shorter input sequence deals with the problem of their quadratic computational complexity, allowing for us to use a larger Transformer model. By comparison, STraTS is a self-supervised transformer model that does not apply any aggregation, and we show that our model outperforms it. Our results in Figure 4 also show that using a more granular input representation with less information loss does not necessarily increase performance, since there is a tradeoff between the granularity and density of inputs.
>
> > The experiments could be improved: a) It would be better if the authors conducted experiments to show which aggregation methods are better for clinical settings (e.g., mean, maximal, minimal, last observed variable)? b) The lengths of different patients’ EHR data vary a lot (e.g., from 1~3 days to several weeks in ICUs). It is tricky to select the value of L for different patients. It would be better if the authors conducted experiments to show how the model is sensitive to L for different groups of patients, especially for patients with extremely short or long observation periods. Moreover, in Figure 4, when L is set 48, the performance becomes worse, which is unusual and worth explaining why. c) The statistics of dataset (e.g., age, gender) are missing. d) It is unclear how to learn patient representation [REP].
>
> Thank you for the useful suggestion for additional experiments to improve our paper. We have added results comparing different aggregation functions in Table 3, additional discussion of our results with different L values in Section 4, and dataset statistics in Appendix C. We would generally treat L as a model hyperparameter that might need to be tuned for different datasets. [REP] is not trained during self-supervised training, so it is learned during task-specific fine-tuning.
>
> > [T]he authors claim that “the use of Transformer in modelling sparse irregular time series with tabular data has not been widely explored.” However, Transformer has been widely used in modeling EHR data (e.g., [1-4]).
>
> Thank you for raising this, we've added additional text in our Introduction and Related Works section to put our work in context with respect to these papers. Med-BERT, BEHRT, and INPREM only use sets of diagnostic codes over time, without numeric inputs, while EhrBERT analyzes the text of EHR notes. By contrast, our model incorporates both discrete and continuous numeric time series data, sampled irregularly over time. Applying self-supervised transformers to this kind of input data is less straightforward, and few papers have been published on it.
>
> [1] Grinsztajn, Léo, Edouard Oyallon, and Gaël Varoquaux. "Why do tree-based models still outperform deep learning on tabular data?." arXiv preprint arXiv:2207.08815 (2022).

---

### Official Review · Reviewer_6eHs · 2022-10-25

**Confidence:** 5
**Correctness:** 2
**Technical Novelty And Significance:** 2
**Empirical Novelty And Significance:** Not applicable
**Recommendation:** 5

**Clarity, Quality, Novelty And Reproducibility:**

**Clarity:** The paper mentions that using continuous value time embeddings where actual time values are taken into account is useful. It is not clear why would this matter as after binning they are just regularly sampled data. Can the authors describe how the baseline approaches were trained? How does the proposed approach differentiates between missing data (from binning) and masked data for pretraining? Are masks used during finetuning stage too? What aggregation function is used, comparison with different aggregation functions?

**Novelty:** The novelty is marginal as the self-supervised training is similar to mTAN and some other approaches mentioned in the paper and the input representation is widely known.

**Reproducibility:** Not reproducible. The paper doesnt include any code or provide hyperparameters required to reproduce the results.


**Strength And Weaknesses:**

### Strengths
1. The paper proposes a simple approach to improve the learning of irregularly sampled time series data using self-supervised pretraining.
2. Experimental results show the effectiveness of the approach when compared to baselines and other recent approaches.
3. The paper focuses on the task of learning from irregularly sampled data which is important in many domains.

### Weaknesses
1. The paper claims to propose the idea of using a binning technique to transform irregularly sampled time series into a regularly sampled sequence data with missing values. This has been widely studied in literature as discussed in this survey ( Section 4.1, [1]).
2. The novelty is marginal as the self-supervised training is similar to mTAN and some other approaches mentioned in the paper and the input representation is widely known.
3. The experimental results seem to be just marginally better than a standard baseline of XGBoost even when the proposed approache uses self-supervised pretraining. XGBoost clearly outperforms the TE(SS-) approach on all datasets. Could this be addressed by pretraining on a large scale time series data? As shown in NLP and vision community, scale of the data really matters. Its something the authors could try to further improve the performance. Since this seems like mostly an empirical work, I would also expect to see some study towards the transfer learning capabilities of the pretrained models.
4. How do the authors deal when different features in the input have different sampling rate which is quite common in healthcare data? How does the proposed approach differentiates between missing data (from binning) and masked data for pretraining?
5. It is not clear if the baseline approaches are trained with input representation mentioned in the current paper or their corresponding papers. Some of the baseline approaches (like mTAND) do not utilize the static inputs. Were these baseline approaches trained with static inputs? mTAND results on PhysioNet seem to be the same as mentioned in the original paper where no static inputs were used.


#### References
1. Shukla, Satya Narayan, and Benjamin M. Marlin. "A survey on principles, models and methods for learning from irregularly sampled time series." arXiv preprint arXiv:2012.00168 (2020).

**Summary Of The Paper:**

The paper focuses on the problem of learning from irregularly sampled time series data. Similar to the NLP and vision community, it employs a self-supervised pretraining approach using Transformers to improve the modeling of irregularly sampled time series data. It first applies a binning technique to covert the irregularly sampled time series to regularly sampled data to apply Transformers models. The paper also proposes self-supervised learning tasks for such data to learn better representation. Experiments show that the proposed approach outperforms several recent approaches on multiple downstream tasks across two datasets.

**Summary Of The Review:**

My recommendation for this paper is weak reject. Although the paper brings some interesting ideas from vision and NLP community, it fails to deliver in terms of the results.

---

> ### Author Response · Authors · 2022-11-17
> **Response to reviewer 6eHs (1/2)**
>
> We thank the reviewer for useful feedback and suggestions. We've clarified a number of points in the paper and added additional experiments, in particular on different aggregation functions. We respond to particular concerns below:
>
> > The paper claims to propose the idea of using a binning technique to transform irregularly sampled time series into a regularly sampled sequence data with missing values. This has been widely studied in literature as discussed in this survey ( Section 4.1, [1]).
>
> Thank you for pointing this out, we've edited the abstract and body of the paper to refine this claim. We recognize that binned input representations have been previously applied for irregularly sampled time series data. However, we believe that our use of binning to enable large transformer models is novel, as previous transformer models in this field have not applied any input aggregation, leading to excessively long input sequences [1,2].
>
> > The experimental results seem to be just marginally better than a standard baseline of XGBoost even when the proposed approache uses self-supervised pretraining. XGBoost clearly outperforms the TE(SS-) approach on all datasets. Could this be addressed by pretraining on a large scale time series data? As shown in NLP and vision community, scale of the data really matters. Its something the authors could try to further improve the performance. Since this seems like mostly an empirical work, I would also expect to see some study towards the transfer learning capabilities of the pretrained models.
>
> Even though our gains over XGBoost aren't large, we believe our model is the first deep learning model to outperform a well-tuned XGBoost baseline on this data, which is an important step in the field (the deep learning baselines we compare to do not include XGBoost in their comparisons, and we significantly outperform them). We do agree that this gain is heavily dependent on self-supervised learning - we don't consider TE(SS-) to be a competitive model, and we only include it to show the impact of SSL. Future work to incorporate even more data in pretraining would be valuable, but it is not straightforward to do with the datasets available to us since they provide different input variables. Our results in Figure 2 do show that our model can learn well when it has extra unlabelled data in addition to the labelled dataset, which is often useful in practical health care settings.
>
> > How do the authors deal when different features in the input have different sampling rate which is quite common in healthcare data? How does the proposed approach differentiates between missing data (from binning) and masked data for pretraining?
>
> The binning approach we use handles irregular sampling by producing a single value for each feature at each time bin. Features which are not sampled in that time bin are treated as missing and are zero-imputed, while multiple samples are combined using an aggregation function. We've added an experiment comparing different aggregation functions in Table 3 in the paper and also shown below, but we find that taking the last sampled value tends to work well.
> |  Aggregation Function | ROC-AUC   | PR-AUC  |
> |---|---|---|
> | Max aggregation  | $0.895 \pm 0.003$  | $0.607 \pm 0.003$  |
> | Mean aggregation | $0.890 \pm 0.005$  | $0.604 \pm 0.007$  |
> | Last observed  | $0.895 \pm 0.002$  |  $0.607 \pm 0.003$ |
>
> Information about missing data and masked data for pretraining are provided to the model in different ways. The tensor $\boldsymbol{m}$ contains binary missingness indicator variables that are passed to the time series input encoder MLP. When a time step is masked for pretraining, the entire encoded input for that time step is replaced by a vector [MASK], which is learned during pre-training. During fine-tuning, [MASK] is not used.
>
> > It is not clear if the baseline approaches are trained with input representation mentioned in the current paper or their corresponding papers. Some of the baseline approaches (like mTAND) do not utilize the static inputs. Were these baseline approaches trained with static inputs? mTAND results on PhysioNet seem to be the same as mentioned in the original paper where no static inputs were used.
>
> Thank you for pointing this out, we have updated Appendix B to ensure that the training procedure, input representations, and static variable usage are clear for all methods. For a fair comparison, we do incorporate static variables in all baselines, using different approaches depending on the model. For mTAND, we represent static variables as time series with only one sample, which matches the mTAN code repository.

---

> > ### Author Response · Authors · 2022-11-17
> > **Response to reviewer 6eHs (2/2)**
> >
> > > The paper mentions that using continuous value time embeddings where actual time values are taken into account is useful. It is not clear why would this matter as after binning they are just regularly sampled data.
> >
> > For tasks such as MIMIC-IV phenotyping where the length of time that each patient is observed can vary, the time periods represented by each time bin can vary from instance to instance. Encoding the actual time values provides additional information to the model in these cases.
> >
> > [1] Sindhu Tipirneni and Chandan K. Reddy. Self-supervised transformer for sparse and irregularly sampled multivariate clinical time-series. ACM Transactions on Knowledge Discovery from Data, 1(1), 2022.
> > [2] Houxing Ren, Jingyuan Wang, Wayne Xin Zhao, and Ning Wu. RAPT: Pre-training of time-aware transformer for learning robust healthcare representation. In ACM SIGKDD Conference on Knowledge Discovery and Data Mining, 2021.

---

### Author Response · Authors · 2022-11-17
**Response to all reviewers**

We would like to thank all of the reviewers for their insightful comments. We are encouraged to see reviews highlighting the simplicity and effectiveness of our proposed self-supervision method as well as strong experimental results shown in the paper (reviewers 6eHs, HAwr, and EBR4). Further, as highlighted by reviewer HAwr, we _study an important problem with a potential for significant impact in using AI for clinical settings_ and the presented model _performs better than XGBoost_. We have made substantial revisions to the paper text and added additional experimental results based on reviewer feedback. Here, we address the general concerns raised in the reviews, and we address reviewers' specific concerns in individual replies.

We would first like to discuss the **novelty** of our contributions. As noted by the reviewers, aspects of our proposed method have appeared in similar forms in various contexts. We nonetheless believe that our work significantly contributes to the field of machine learning for EHR data because using transformers and self-supervision for ICU data poses many challenges, and our design decisions around input representations and self-supervised training address these challenges in substantially different ways from previous methods. While Med-BERT[1], BEHRT[2], and INPREM[3] use transformers, they do not use numeric time series data at all, limiting their inputs to sets of diagnostic codes applied over time. Indeed, the problem of using self-supervised transformers for sparse *numeric* EHR data is not well-explored in the literature. STraTS is the closest work to our proposed model, but is limited to using a smaller transformer due to its choice for representation for long input sequences. We demonstrate that input binning improves transformer performance, since the potential loss of input granularity is outweighed by the ability to train deeper models, and our additions of presence prediction and masked event dropout increase the efficacy of self-supervised training.

We would also like to address concerns regarding the **strength of experimental results**. Previous works we compare our model against do not generally compare results to a well-tuned XGBoost model. However, this is a crucial baseline to compare against (see https://arxiv.org/abs/2207.08815). We show that (a) XGBoost outperforms state-of-the-art deep learning methods on numeric EHR data and (b) our model beats XGBoost on MIMIC and PhysioNet mortality tasks and matches its performance on MIMIC phenotyping; although the gains may be modest, we believe these results are an important milestone for futher advances in this field. There are important advantages to having a deep learning method that outperforms XGBoost in this domain: first, they permit learning from unlabelled data or from other datasets, the advantages of which can be seen in our experiments on limited labelled data. Second, they can be extended to handle multimodal inputs, which can often be harder for XGBoost to incorporate. Our work can pave the way for future models to further increase the performance gap over XGBoost.

The reviewers have also raised the concern of **reproducibility**. To ensure our results are reproducible, we will be publishing implementation code for our model as well as the IDs for the patient-level splits on our lab's Github page along with the final paper. We have also presented all of our model's hyperparameters in the paper, with those not mentioned in the main text being specified in Appendix A. We have added a reproducibility statement to the paper describing these steps.

To facilitate reviews, we have included a PDF in the Supplementary Material showing the changes made to the text of the paper since our original submission. Please note that not all changes to tables and figures are highlighted in this file.

[1] Rasmy L, Xiang Y, Xie Z, et al. Med-BERT: pretrained contextualized embeddings on large-scale structured electronic health records for disease prediction. NPJ digital medicine, 2021, 4(1): 1-13.

[2] Li Y, Rao S, Solares J R A, et al. BEHRT: transformer for electronic health records. Scientific reports, 2020, 10(1): 1-12.

[3] Zhang X, Qian B, Cao S, et al. INPREM: an interpretable and trustworthy predictive model for healthcare//Proceedings of the 26th ACM SIGKDD International Conference on Knowledge Discovery & Data Mining. 2020: 450-460.

---

### Decision · Program_Chairs · 2023-01-20

**Decision:**

Reject

**Justification For Why Not Higher Score:**

Method somewhat different to prior work, but experimental results show that the resulting models aren't actually all that different in terms of performance.

**Justification For Why Not Lower Score:**

N/A

**Metareview: Summary, Strengths And Weaknesses:**

The paper proposes a new transformer-based method that handles sparse data, and the experiments show that it is possible to compress EHR data and preserve information. The proposed method is demonstrated effective on MIMIC and PhysioNet. All reviewers pointed out a lack of novelty in the proposed method. The authors pointed out differences between their work and those cited by the reviewers. The question is whether these differences are sufficient to warrant publication of a new technique, especially given the large number of time series analysis methods that already exist. To ascertain this, I considered the experimental results - the idea is that if the method brings considerable performance gains, then it clearly exploits some aspect of the data that previous methods did not. Unfortunately, looking at Table 1, the gains are marginal. It is true that for this type of data, it's difficult to obtain vastly superior results, and that the stdevs are small as well, but, event so, improvements of less than 0.01 and 0.02, respectively, in AUC scores are underwhelming. The numbers reported in the STraTS paper (i.e. previous sota) for PhysioNet are different, though this could be due to the experimental setting. The STraTS paper itself only showed marginal improvements over the baseline, but it was not published in a venue like ICLR/ICML/NeurIPS.

**Summary Of Ac-Reviewer Meeting:**

N/A